# Encryption Scheme of Verifiable Search Based on Blockchain in Cloud Environment

**Buzhen He**  **and Tao Feng** *

School of Computer and Communication, Lanzhou University of Technology, Lanzhou 730050, China
* Correspondence: fengt@lut.edu.cn

**Abstract:** While transferring data to cloud servers frees users from having to manage it, it eventually raises new problems, such as data privacy. The concept of searchable encryption has drawn more and more focus in research as a means of resolving the tension between data accessibility and data privacy. Due to the lack of integrity and correctness authentication in most searchable encryption techniques, malicious cloud servers may deliver false search results to users. Based on public key encryption with searching (PEKS), the study suggests a privacy-preserving method for verifiable fuzzy keyword searches based on the Ethernet blockchain in a cloud context to overcome the aforementioned security concerns. The search user can check the accuracy and integrity of the query document using the unalterability characteristics of the Ethernet blockchain system in this scheme to prevent the cloud server from giving incorrect query results. Furthermore, a fair transaction between the cloud server and the data user is achieved and can be tracked back to the malicious user using hash functions and Ethereum smart contracts, even if the user or the cloud is malicious. Finally, the security analysis shows that, under the random oracle model, our technique fulfils the adaptive selection keyword's semantic security. The performance assessment demonstrates that the proposed scheme outperforms other related schemes in terms of computational efficiency.

**Keywords:** searchable encryption; fuzzy keyword search; blockchain; verifiable; fair payment

## 1. Introduction

Cloud storage has gained popularity due to its low cost, high power, and low-cost benefits, but there are significant security risks associated with its use, such as malicious use of system or network vulnerabilities and other methods to steal or tamper with user data during data transmission, resulting in the leakage of private information, so users must carefully consider these risks before using cloud storage.

At the same time, cloud service providers are not completely trustworthy. Companies storing data on cloud servers run the risk of having their information leaked to users, including rivals; people storing sensitive information on cloud servers run the risk of having their privacy violated; and malicious cloud servers run the risk of deleting files that users have not used in a while and compromising data integrity. Additionally, some cloud service providers will conceal their data leaks as much as they can to preserve their good reputation and avoid repercussions.

Most users choose to encrypt file contents before uploading data operations in order to prevent privacy leaks from cloud service providers and to stop private file contents from being revealed during the search process. To resolve the aforementioned issue, the searchable encryption (SE) approach for keywords is required. Searchable encryption is a crucial method that may easily search the data in the cloud storage without downloading the entire ciphertext document and extracting the ciphertext directly. Schemes for a single keyword query [1,2], multi-keyword query [3,4], fuzzy keyword query [5,6], and sorted keyword query [7,8] searches have all been proposed for searchable encryption.

Cloud servers that are "semi-honest and curious" can only complete a portion of the search operation and provide inaccurate search results [9]. Verifiable search encryption

techniques are suggested as a solution to these issues, where the ciphertext must be delivered to the user only after being confirmed to be accurate and authentic. Additionally, the majority of customers enter into a payment agreement with the cloud service provider before using the search function; thus, the cloud server can only be paid for the outsourced computation when the user obtains accurate and comprehensive search results. In addition, if the payment model is adjusted to pay when results are received, there may be cases where malicious users refuse to pay for the service even when they obtain the right results.

Ethernet is a public blockchain platform with smart contract functionality, and its smart contracts [10] allow it to enable trusted transactions between anonymous parties without a central authority. Therefore, smart contracts are more suitable for searchable cryptosystems.

The concerns of data privacy leakage, unverifiability, insecure transmissions, untraceable harmful users, and unfair payments are all addressed by the blockchain-based verifiable search encryption method that we present in this work. The following are the primary research topics covered in this essay:

(1) The paper proposes a verifiable fuzzy keyword search encryption scheme based on blockchain in a cloud environment. Users enter keywords, and the system provides the document data that most closely matches them. The searchable encryption system, which accomplishes fair payment, maintains the dependability and credibility of the scheme and has superior security and efficiency, verifying the accuracy and integrity of the search results;

(2) In order to achieve the traceability of malevolent users or unfair transaction information, the user's identity information and transaction records are saved on the blockchain after the transaction is complete;

(3) The security analysis demonstrates that this strategy successfully protects data privacy from adaptive selection keyword attacks while maintaining the confidentiality of encrypted data.

The remainder of this paper is organised as follows. We present related work in Section 2 and briefly describe some scenario models in Section 3, which include a system model, a threat model, and a security model. We give a concrete scheme in Section 4, which is divided into eight steps and described in detail. We demonstrate the security of the proposed solution in Section 5, as well as a functional comparison and performance comparison with other solutions. We draw the concluding remarks in Section 6.

## 2. Related Work

A keyword trapdoor was built to compare with each ciphertext keyword in the ciphertext document in order to achieve a single-keyword search of encrypted data, as proposed by Song et al. in their notion of searchable encryption [11]. However, retrieval based on a single keyword only yields a huge number of relevant articles, and numerous multi-keyword retrieval research approaches have been developed. The first searchable encryption technique based on concatenated keywords was presented by Golle et al. [12]; nevertheless, the retrieval efficiency of the scheme is low, making it less useful. Goh [13] proposed to build an index for each outsourced document and use the index to complete the retrieval without matching each document one by one, creating a searchable encryption scheme based on orthogonal indexing. While the sorted, searchable encryption scheme can return the top k search results related to the keywords and has achieved many research results in recent years, Goh's proposal can return the retrieval without matching each document one by one. Zhang et al. [14] designed a searchable encryption scheme combining keyword weights and a two-factor ranking function of keyword similarity to rank the search results and improve the usability of the scheme. Even when consumers enter terms with a few minor spelling mistakes, the fuzzy query makes it easy to receive pertinent results. The literature [15] suggests a privacy-preserving approach for fuzzy multi-keyword searches using cloud services, which addresses the issue of the previous schemes' poor performance and noticeably raises search efficiency and matching precision.

The majority of these cloud-based searchable encryption techniques take into account honest and inquisitive server architectures. Verifying the retrieval results is necessary since, in practice, cloud servers may give users partial or inaccurate search results. Verifiable privacy-preserving search strategies have thus become a popular study area. The first verified search encryption method (VSSE), which offers consumers data privacy as well as query correctness and integrity, was proposed by Chai et al. [16] in 2012. A verified search system was proposed by Kurosawa et al. [17], but it has the disadvantage of a high verification overhead and cannot verify whether the returned results have been updated or deleted. Li et al. [18] used Paillier homomorphic encryption and the Key Hash Message Authentication Code (HMAC) to confirm the accuracy and integrity of the encrypted search results. A multi-keyword verification-enabled attribute-based encryption approach was suggested in the literature [19], adding a third-party entity to the scheme and utilizing a testing mechanism to stop unreliable cloud servers from generating inaccurate search results.

The above verifiable search encryption algorithms, however, lack a review mechanism that can be used for all search schemes; thus, blockchain and smart contracts are used to guarantee fairness for each participant. The literature [4] developed a blockchain-based multi-keyword sorted search and fair payment system in the blockchain-based verified scheme, returning accurate and comprehensive search results to the data users. In the literature [20], a search index was created using bitmaps, which increased search efficiency and enabled blockchain to verify the accuracy of the search results. In order to provide end users with privacy-preserving and verifiable query functionalities in industrial IoT systems, the literature [21] presents a blockchain-based query verification model enabling multiple signatures.

Table 1 lists the main contributions of selected literature and their respective limitations. Therefore, we provide a blockchain-based verifiable search encryption technique for cloud services in this study and an improved existing verifiable searchable encryption scheme. To ensure the traceability of the identity information of the data user, this system uses a one-to-many search model and stores the identity information in the blockchain each time a search request is made by a data user. Additionally, it makes use of a smart contract to confirm the accuracy and reliability of the papers that are returned in order to guarantee that consumers only pay after receiving the proper results and stores the transaction data in the blockchain after the transaction is complete.

**Table 1.** Comparison of related works.

| Literature | Year | Main Contribution | Drawbacks |
|:---:|:---:|:---:|:---:|
| [22] | 2016 | Dynamic fuzzy verifiable search scheme | Low verification efficiency |
| [23] | 2018 | Propose a form of "deposit" in the blockchain | Large number of signature verification computations |
| [24] | 2019 | Multiple users, high search efficiency | The authorization issue is not addressed |
| [25] | 2020 | Proposed a dynamic single sign-on solution based on blockchain | High workload and inefficient verification |
| [26] | 2020 | Verification algorithms are added to the decryption process in this approach. | Problems between users and the cloud service platform cannot be resolved. |
| [4] | 2021 | Proposed a blockchain-enabled scheme with multi-keyword search (BPKEMS) | Users are inefficient when they make spelling mistakes |
| [20] | 2022 | Improved search efficiency by using bitmaps | Only store a small quantity of information |
| [21] | 2021 | Provide privacy protection and verifiable query capabilities for end users in IoT (Internet of Things) systems | No security analysis of the scheme |

## 3. Scheme Model

This section provides an overview of a blockchain-based verifiable search encryption technique used in a cloud context, including the system model, algorithm specification, threat model, and security model.

### 3.1. The System Model

Figure 1 depicts the system model for the verifiable search encryption scheme based on blockchain in the cloud environment that is suggested in this paper. The key is distributed by the trusted authorization; the data owner encrypts the documents and sends them to the cloud server, the data user submits a search request, the cloud server completes the search, the blockchain completes the document's integrity and correctness verification, as well as the data user's authentication, and then it returns the correct encrypted document that has passed the verification to the data user. Participants can be categorized into one of five categories: semi-convertible cloud servers (SCS), data owners (DO), data users (DU), blockchain (BC), and trusted authorization (TA).

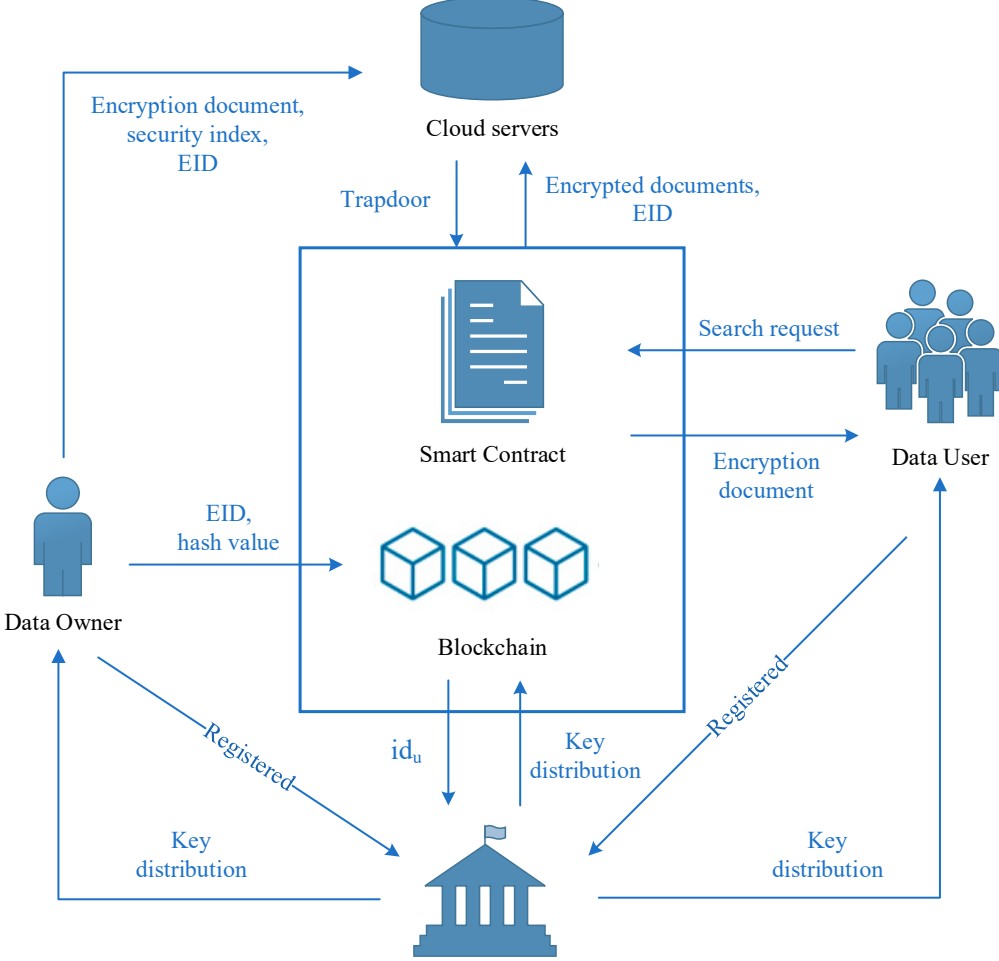

**Figure 1.** The system model.

The following is a description of each entity's role and the functions to which they belong:

Trusted authorization. The trusted authorization is responsible for generating public and private keys for each user and publishing the public parameters of the system.

Cloud servers. Cloud servers with strong processing capacity and storage space but a dubious reputation will exist to attempt collecting users' personal information. In this case, the cloud server is primarily in charge of keeping the encrypted documents that the

data owner has uploaded and carrying out search operations to deliver the most relevant documents to the data user.

Data owner. The data owner is a user who shares data with other users and owns the original data, $D = \{D_1, D_2, D_3, \cdots, D_n\}$. The main work is to compute the secure index and encrypt the documents and upload them to the cloud server for storage.

Blockchain technology. In order to identify the data user who acquired this encrypted document, it is primarily responsible for documenting the identity ID of the data user and associated transactions. Additionally, it establishes an environment for fair payment from both sides by comparing the hash value to the integrity and correctness of the encrypted content.

Data users. Data users are users who have a search requirement, construct a search token by encrypting the terms they wish to use, and transmit it to the smart contract. They then wait to receive the appropriate document, which has been confirmed by the smart contract.

### 3.2. Threat Model and Security Model

The threat model in this instance is as follows. Since the cloud server is honest and curious, it tries to find out about the user's private information and make assumptions about it, risking the user's security and privacy. The blockchain is totally trusted to store the data user's $id_u$ and to verify the validity and reliability of the results that have been returned. Public and private keys are generated for each user by a completely trusted trust center, and the system's public parameters are made available to the public.

The security model in this instance is as follows. In this paper, we use the security model proposed in the literature [13], which was first proposed by Goh as Indistinguishability under Chosen Keyword Attack (IND-CKA). From the proof equivalence of the literature [16], it follows that a simulation-based proof of a game is equivalent to a proof of an indistinguishable game, where the adversary, $A$, will win by analysing the ciphertext generated by the simulator, the index, and the distinguishability of the search token game, and the CSP and external attackers will not obtain any additional information beyond the search pattern. Implementing IND-CKA security means that adversary $A$ cannot infer the contents of documents, indexes, and search tokens from the leaked content. The game will be described below.

A simulation-based game between attacker $A$ and simulator $S$ is used to prove the security of the scheme while allowing the leakage of search patterns. Two leakage functions are used to represent the information leakage scenario of this method, $L = (L_1, L_2)$, $L_1$ is defined as $L_1(D) = (C'_R)$, input plaintext document set, output ciphertext document $C'_R$, $L_2$ is defined as $L_2(D, w') = (F', \pi', T'_{w'})$, input document collection and keywords, output search token $T'_{w'}$ and function $F', \pi'$. The game played between challenger $C$, adversary $A$, and simulator $S$ is defined as follows.

Define the following games:

$Real_A^{\Pi}(\lambda)$: Completed by the contestant and foe, $A$. The initialization algorithm, $Setup(1^\lambda)$, is executed by the game. Challenger $C$ receives the document set, $D$, from adversary $A$. Adversary $A$ receives the result $(C_R, I)$ after the challenger computes the encrypted document set, $C_R$, and the security index, $I$. The antagonist chooses a keyword, $w'$, at random and sends it to the challenger. The challenger computes and generates the relevant search token, $T_{w'}$, to send to the adversary during the search phase. The opponent eventually outputs bit, $b$.

$Ideal_{A,S}^{\Pi}(\lambda)$: Executed by simulator $S$. The challenger receives the document set, $D$, after adversary $A$ chooses it. Based on the leakage functions, $L_1$ and $L_2$, $S$ computes $\left(C'_R, I'\right)$ and transmits it to the challenger. Simulator $S$ computes the relevant search token, $T'_{w'}$, together with the pseudo-random function, $PRF\ F$, and pseudo-random permutation function, $PRP\ \pi$, based on the leakage function, $L_2$, and sends it to the challenger after A chooses the keyword, $w'$, and delivers it to them. The challenger sends the search token, $T'_{w'}$, to adversary $A$. The adversary then outputs bit, $b$.

## 4. Specific Structure

Table 2 displays some of the symbols used in this scheme along with the descriptions that go with them. In this section, we describe the scheme in eight steps, as follows.

**Table 2.** Symbol and meaning.

| Symbol | Meaning |
| --- | --- |
| $D$ | Collection of plaintext documents |
| $C_R$ | Collection of ciphertext documents |
| $W$ | Keyword collection extracted from $D$ |
| $EID$ | Encrypted document identifier set |
| $N$ | Document number |
| $I$ | Secure index |
| $K_1, K_2, K_3$ | Encryption cipher and trapdoor |
| $S_{w_i,d}$ | Fuzzy word set |
| $T_{w'}$ | Search token of keyword $w'$ |
| $TD_{w'}$ | Trapdoor for $w'$ |
| $C_{R_{w'}}$ | Ciphertext set containing search keywords |
| $C_T$ | The ciphertext, after passing authentication |

### 4.1. Algorithm Defined

The article suggests the following eight polynomial-time encryption algorithms as part of a blockchain-based, searchable, verifiable encryption system for cloud environments.

$$\pi = (Setup, UserRegist, Enc, IndexGen, TokenGen, Search, Verify, Dec)$$

**Initialization algorithm** $Setup(1^\lambda) \to param, msk$: the system's public parameter, *param*, and the system's private key, *msk*, are output by the trusted authorization after the security parameter is submitted.

**Registration algorithm** $UserRegist(param, msk, id_O, id_U) \to PK_O, PK_U, psk_O, psk_U$: In order to seek registration, the data owner sends the trust center their identification information. The trust center receives this information and uses it to input the public parameter, param, and the system's private keys, *msk* and $id_O$, and to output a public key, $PK_O$, and partial private key, $psk_O$, in response. Similar circumstances arise when a data user requests registration.

**Encryption algorithms** $Enc(param, D, K_1, K_2) \to C_R$: to create the ciphertext document, $C_R$, the data owner, *DO*, uses $K_1, K_2$ to encrypt the plaintext document, $D_i(i \in [n])$.

**Secure index generation algorithm** $IndexGen(param, D, K_1, K_2) \to I$: the *DO* builds the index by scanning the plaintext to receive the keyword set and encrypt it with the keys $K_1, K_2$ to receive the secure index, $I$.

**Search token generation algorithm** $TokenGen(param, w', id_U, K_1, K_3) \to TD, T_{w'}$: The data user, *DU*, enters the keyword, $w'$, to be searched for and the key, $K_1$, to create the trapdoor, *TD*. Based on the trapdoor, *TD*, $K_3$, and the user identity code, $id_u$, the search token, $T_{w'}$, is then created and sent to the smart contract.

**Search algorithm** $Search(I, TD, C) \to C_T, EID$: The cloud server runs the search operation to compare the trapdoor, *TD*, with the security index, $I$. The ciphertext, $C_T$, containing the keyword and its matching identifier, *EID*, is acquired if the match is successful.

**Verification algorithm** $Verify(C_{Rw'}, H_O) \to n$: The smart contract calculates the hash value, $H_N$, after receiving the encrypted document, compares it to the hash value of the matching document stored in the blockchain, and outputs $n = 1$ if $H_N = H_O$; otherwise, $n = 0$.

**Decryption algorithm** $Dec(C_T, K_2) \to D_T$: The user receives the ciphertext result, $C_T$, given by the blockchain after paying the service charge, which they then decrypt using the symmetric key, $K_2$, to obtain the proper plaintext document, $D_T$.

### 4.2. Content Initialization Phase

To create the master key and system parameters, the trusted authorization follows these procedures.

- The trusted authorization takes the security parameter and creates two multiplicative cyclic groups, $G_1, G_2$ on $\mathbb{Z}_p$, where $p$ is a large prime, and $g$ is the generating element of $G_1$. $e : G_1 \times G_1 \to G_2$ is a bilinear mapping;
- Randomly select $s \in Z_p{}^*$ and calculate $P = g^s$;
- Select two collision-resistant hash functions:

$$H_1 : \{0,1\}^* \to G_1$$

$$H_2 : \{0,1\}^* \to Z_p{}^*$$

- Take a pseudo-random function, *PRF F*, and a pseudo-random permutation function, *PRP* $\pi$, with the following parameters:

$$F : \{0,1\}^k \times \{0,1\}^l \to \{0,1\}^N$$

$$\pi : \{0,1\}^k \times \{0,1\}^l \to \{0,1\}^l$$

Output the system's public parameter, $param = (G_1, G_2, p, \mathrm{e}, g, P, H_1, H_2, F, \pi)$ and the system's master key, $msk = e< g, g >^s$.

### 4.3. Key Generation Phase

The *DO* randomly selects $x_O \in Z_p{}^*$, calculates $id_O$, and sends it to the trusted authorization, which calculates the *DO*'s partial key, $psk_O = H_1(id_O)^{msk}$, and public key, $PK_O = g^{id_O}$. A part of the private key is sent to the data owner through the secure channel, and the data owner calculates the private key, $SK_O = H_2\left(x_O, H_1(id_O)^{msk}\right)$. Similarly, the *DU* obtains the public key, $PK_U = g^{id_U}$, and the private key, $SK_U = H_2\left(x_U, H_1(id_U)^{msk}\right)$. The trustworthy center uses $PK_O$ to encrypt $K_1, K_2$ to send to the data owner and $PK_U$ to encrypt $K_1, K_3$ to send to the data user.

### 4.4. Index Building, Ciphertext Encryption Phase

The inverted index method is used to construct the index table. By scanning the plaintext document, the set of keywords, $W = \{w_1, w_2, \dots, w_m\}$, is initially extracted, and the n-dimensional index vector of each keyword, $w_i$, is indicated by $v(w_i)$. If the keyword appears in the document, $v(w_i)[j] = 1$; otherwise, $v(w_i)[j] = 0$.

A fuzzy set, $S_{w_i,d} = \{w_{i,1}, w_{i,2}, \dots, w_{i,t}\}$, is constructed to represent the fuzzy set with the keyword, $w_i$, and editing distance, $d$, where $w_{i,t} \left(1 \le i \le n, 1 \le t \le \left|S_{w_i,d}\right|\right)$ denotes the $t$ keyword in the fuzzy set.

For each keyword, $w_{i,t}$, in the fuzzy set, $S_{w_i,d}$, the pseudo-random permutation function, *PRP* $\pi$, is used to confuse the real position of the keyword to obtain $\pi_{k_1}(w_{i,t})$. The pseudo-random function, *PRF F*, is used to calculate $Ev(w_i) \leftarrow F_{k_2}\left(\pi_{k_1}(w_i)\right) \bigoplus v(w_i)$, and the encrypted index vector, $Ev(w_i)$, is obtained. $\pi_{k_1}(w_{i,t})$ is stored on the first node of the inverted index, and $Ev(w_i)$ is stored on the second node to construct the security index, $I$, as shown in Figure 2.

For example, for plaintext documents, $D_1 = I\ am\ happy\ everyday$ and $D_2 = I\ am\ a\ good\ person$. For extracting a collection of keywords, $W = \{w_1, w_2, w_3, w_4, w_5, w_6, w_7\} = \{I, am, happy, everyday, a, good, person\}$.

Then, the keywords $am, a, good$ can be constructed as 2-dimensional vectors, $v(am) = [11]$, $v(a) = [01]$, and $v(good) = [01]$, respectively.

For constructing fuzzy keyword sets, $S_{am,1} = \{am, *am, *m, a*, am*\}$, $S_{a,1} = \{a, *a, *, a*\}, S_{good,1} = \{good, *good, *ood, g*od, go*d, goo*, good*\}$ with an

edit distance of 1 for the keywords *am*, *a*, *good*, respectively. Furthermore, using the pseudo-random function, *PRP* $\pi$, for each keyword in the fuzzy set, gives $\pi_{k_1}(am), \pi_{k_1}(* am), \cdots\cdots$ Then, store it on the first node.

Then, using the pseudo-random function, *PRF F*, the encrypted index vectors $Ev(am)$, $Ev(a), Ev(good)$ are computed and stored on the second node. The resulting security index is then shown in Figure 3.

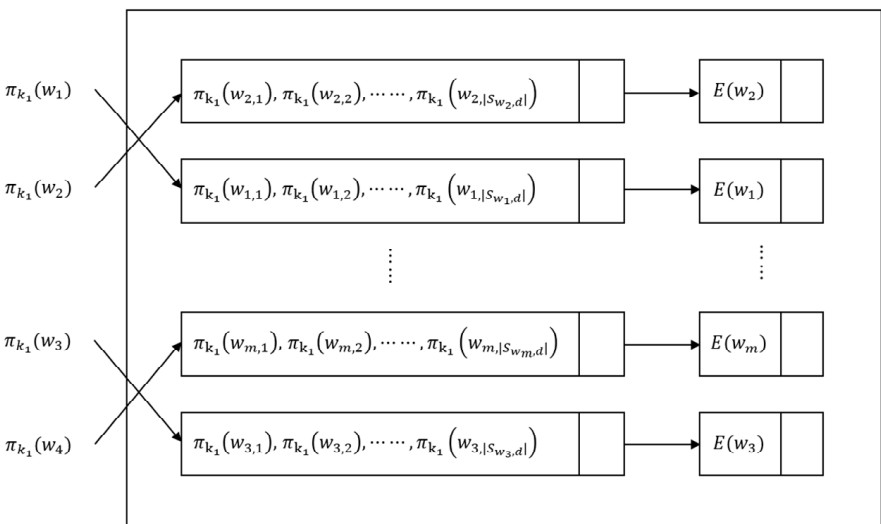

**Figure 2.** The structure of the secure index.

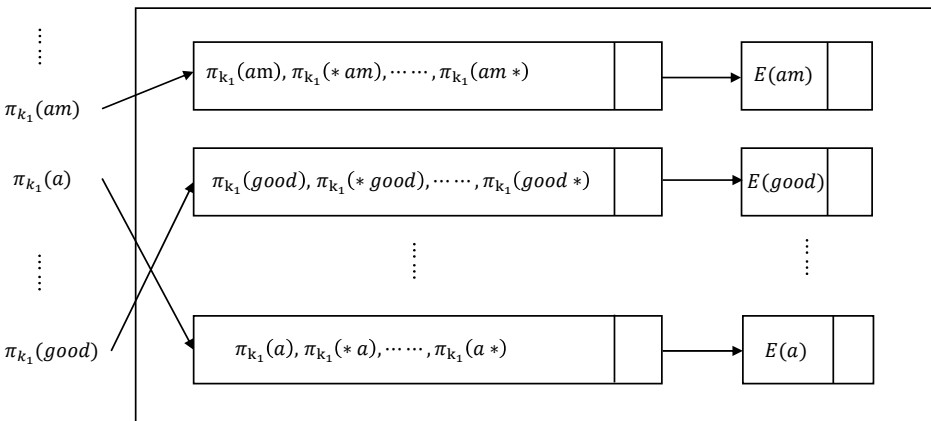

**Figure 3.** Example of a secure index.

For encryption with symmetric keys, $Enc(param, D, K_1, K_3) \to C_R, EID$. To obtain the ciphertext $C_i(i \in [1, n])$ and the encrypted document number $N_i = \{n_i | i \in [1, n]\}$, the *DO* encrypts the document, $D_i(i \in [1, n])$, using $K_1$. To encrypt the document identifier set, $EID = \{EID_1, EID_2, \ldots, EID_n\}$, you must first calculate the hash value, $H_O = H_2(C_i || N_i), i \in [1, n]$, pack the ciphertext set , *C*, and document number , *N*, and then use the key, $K_2$, to encrypt to retrieve the ciphertext $C_R$.

Send the encrypted document identifier, *EID*, and hash result, $H_O$, to the blockchain for storage to make it easier to conduct the ensuing verification operation. Send the ciphertext, $C_R$, security index,*I*, and encrypted document identifier, *EID*, to the cloud server for the search operation.

### 4.5. The Search Token Generation Phase

A search token is created when a user wishes to look up a keyword, and a deposit is needed to stop the user from backing out of the transaction in the middle of it. The user

types in the keyword $w'$ and then uses the keys $K_1, K_3$ to construct the search trapdoor, $TD_{w'} = \left(\pi_{k_1}(w'), F_{k_3}\left(\pi_{k_1}(w')\right)\right)$, and then combines the trapdoor with the user identity code, $id_U$, to create the search token $T_{w'} = (TD_{w'}, id_U)$, which is transmitted to the search.

*4.6. Search Phase*

The search phase is divided into the following three steps:

1.  Verification of identity. The user identity code, $id_U$, is saved to the blockchain after the smart contract receives the user's search request and sends it to the trusted institution to be verified as the user's identity. Once the verification is legal, the keys $K_2, K_3$ are sent to the smart contract through a secure channel, and the user identity code, $id_U$, is saved to the blockchain, where the identity information of the malicious user can be traced using the blockchain's tamper-evident feature;

2.  Search for documents. The smart contract sends the trapdoor, $TD_{w'} = \pi_{k_1}(w')$, to the cloud server, and the cloud server pays the search deposit for the search operation. Compare $\pi_{k_1}(w')$ with the first element, $\pi_{k_1}(w_{i,1})$, of each linked list in the list and then match the other encryption keywords, $\pi_{k_1}(w_{i,t})$; calculate $v(w') \leftarrow F_{k_2}\left(\pi_{k_1}(w')\right) \bigoplus Ev(w')$ to obtain the index vector, $v(w')$. If $v(w')[j] = 1$, add this ciphertext to ciphertext set $C_{Rw'}$ to obtain ciphertext set $C_{Rw'}$, containing search keywords.

3.  Finally, the blockchain receives the ciphertext set, $C_{Rw'}$, and its matching encrypted document identification, $EID_{w'}$.

The detailed process of the search is shown in Algorithm 1.

---

**Algorithm 1** Search

---

**Input:**
The trapdoor, **TD**, the secure index, **I**, and the collection of all cipher documents, **C**;
**Output:**
The ciphertext, $C_{Rw'}$ with the keyword and its matching encrypted document identification, $EID_{w'}$.

1:    Resolution of **TD** into $(\alpha, \beta)$
2:    for $i \leftarrow 1$ to n do
3:    for $t \leftarrow 1$ to $\left|S_{w_i, d}\right|$ do
4:    if $\pi_{k_1}(w_{i,t}) = \alpha$ and $t = 1$ then
5:    Obtain the corresponding second node in which the stored $Ev(w_i)$
6:    Decrypting it to $v(w_i)$ using $\beta$
7:    for $j \leftarrow 1$ to N do
8:    if $v(w_i)[j] = 1$ then
9:    Add $C_j$ to $C_{Rw'}$, $EID_j$ to $EID_{w'}$
10:    end if
11:    end for
12:    return $C_{Rw'}$
13:    end if
14:    if $\pi_{k_1}(w_{i,t}) = \alpha$ and $t \neq 1$ then
15:    Computes $F_{k_2}\left(\pi_{k_1}(w_i)\right)$
16:    Obtain the corresponding second node in which the stored $Ev(w_i)$
17:    for $a \leftarrow 1$ to N do
18:    if $v(w_a)[a] = 1$ then
19:    Add $C_a$ to $C_{Rw'}$, $EID_j$ to $EID_{w'}$
20:    end if
21:    end for
22:    return $C_{Rw'}$
23:    end if
24:    end for
25:    end for

---

### 4.7. Validation Phase

The smart contract receives the hash value, $H_O$, from the data owner in accordance with the blockchain search, $EID_{w'}$. The smart contract determines $H_N = H_2(C_T||N_T)$ by decrypting the ciphertext set, $C_{Rw'}$, with the key, $K_2$. It then retrieves the ciphertext, $C_T$, and the document number, $N_T$. We may determine whether $H_O$ and $H_N$ are equal by comparing the $H_O$ stored in the blockchain. The encrypted document, $C$, is provided to the data user, who then pays the service fee, and the transaction is successful if $H_O = H_N$, indicating that the server returned the proper result. If $H_O \neq H_N$, the transaction is abandoned because it suggests a malicious server that does not return the right document. The method is displayed in Figure 4.

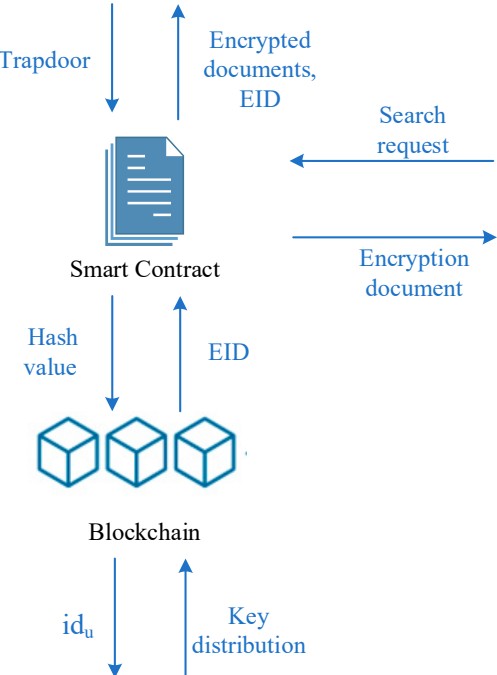

**Figure 4.** Verification process.

The detailed procedure for verification is shown in Algorithm 2.

---

**Algorithm 2** Result Verification

---

**Input:** The ciphertext collection, $C_{Rw'}$ obtained from the cloud search, $H_O$.
**Output:** Verification result, $n$.

1:    Decrypt the ciphertext collection, $C_{Rw'}$ to obtain the ciphertext collection, $C_T$, and the document number collection, $N_T$.
2:    $n \leftarrow 1$
3:    for $i \leftarrow 1$ to $m$ do
4:    $H_{N_i} \leftarrow H_3(C_{T_i}||N_{T_i})$
5:    if $H_{O_i} = H_{N_i}$ then
6:    $n' \leftarrow 1$
7:    else if $H_{O_i} \neq H_{N_i}$ then
8:    $n' \leftarrow 0$
9:    end if
10:    if $n' = 0$ then
11:    $n \leftarrow 0$
12:    break
13:    end if
14:    end for

---

Finally, if the transaction is lawful and the verification results indicate that it is, the user will pay the server the service fee, and the deposits made by both parties will be reimbursed. If the transaction is illegal, the user will receive their deposit back from both parties without having to pay a service charge.

### 4.8. User Decryption Phase

Following the delivery of the service charge, the user receives the correct encrypted document, $C_T$, and uses the key, $K_1$, to decode the plaintext document, $D_T = Dec(K_1, C_T)$. The collection of papers, called $D_T$, that the data user requested has the correct keywords or those that are the closest to them. The encryption scheme timing of a verifiable search based on blockchain in the cloud environment is shown in Figure 5.

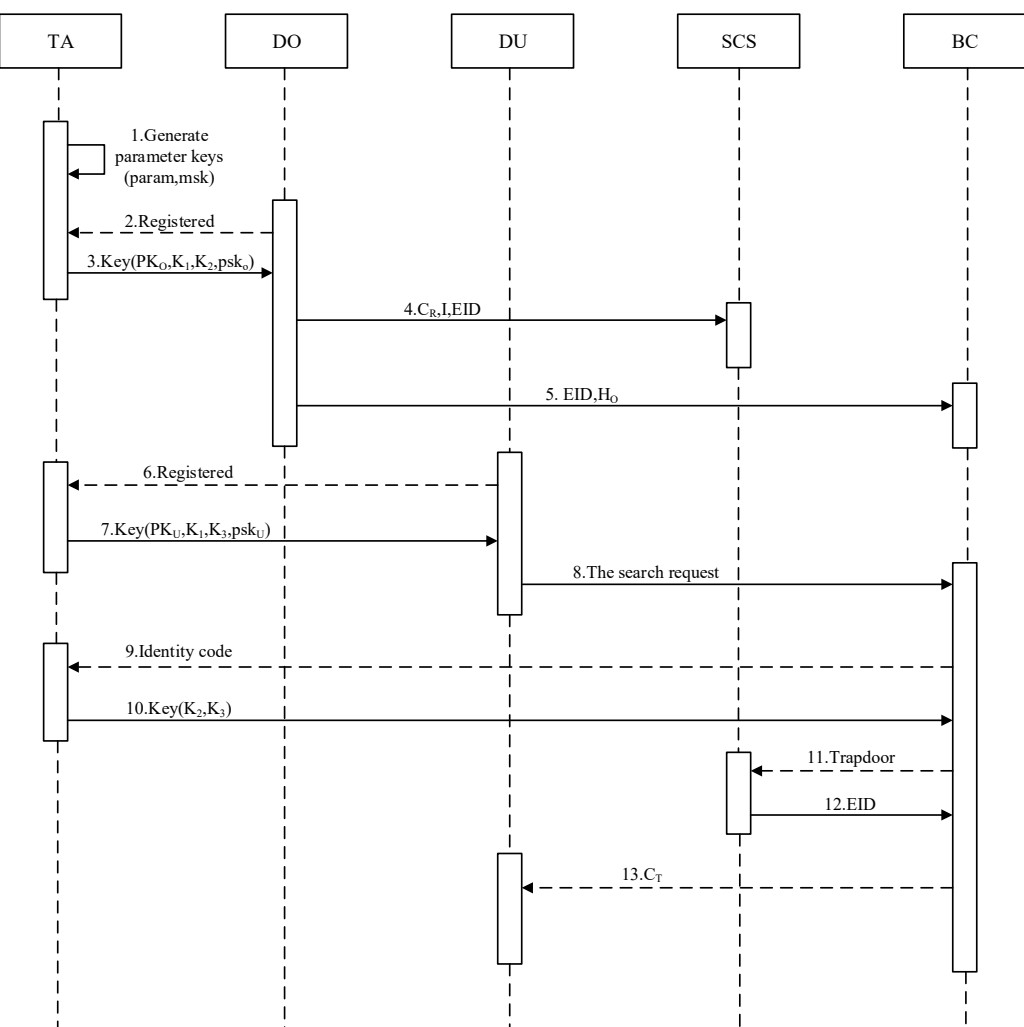

**Figure 5.** Sequence diagram of the scheme.

## 5. Performance Analysis

In this section, we analyse our work in terms of privacy, verifiability, traceability, fairness, and security. We also compare our solution with other similar solutions in terms of functionality and performance.

### 5.1. Security Analysis

The proposed plan can accomplish the following security goals:

**Privacy.** No one can acquire plaintext or keyword information without a key when storing the data in this manner. Furthermore, the key is only available to participants who

have been approved by the reliable centre. The block chain will keep track of the user identifying the information once the user makes the search request, authenticate its validity, and ensure that the user's privacy is preserved before returning the ciphertext to the user.

**Verifiable and traceability.** Using blockchain technology, this strategy Because the transaction records are stored in the blockchain, which is unchangeable and traceable, there is no malicious tampering of the results, and it is convenient to find the user information involved in the transaction at that time when there is an objection to the results. This ensures that data users get the best matching results they need.

**Fairness.** The scheme introduces the Ethereum trading mechanism. When the smart contract determines that the result returned by the cloud server is correct, the user will pay the service fee after refunding the deposit of both parties; otherwise, the deposit of the cloud server will be directly forwarded to the data user. Similarly, only after the user pays the service fee can he obtain the correct return result, and he cannot quit the transaction halfway; otherwise, he will not return the deposit. Transactions are stored openly and transparently in the blockchain, and users can view the transaction information at any time to realize fair and just payment.

**Security.** Under the random oracle model, our technique fulfils the adaptive selection keyword's semantic security (a full proof is given in Appendix A).

*5.2. Functional Comparison*

By comparing the existing searchable encryption schemes, we can find that the scheme in reference [27] implements a searchable encryption scheme that supports sorting but does not support the verification of the results. The literature of [28] can verify the correctness and integrity of the final results of documents, but it is not based on blockchain but checked by data users, which cannot avoid the failure of malicious verification by data users. The literature of [29] realizes verification on the blockchain, which ensures fairness. However, instead of searching based on the cloud server, a smart contract is used, and its search efficiency is far less than that of the former. Document [25] realizes the search based on the cloud server but does not realize the verification and traceability of the user identity. In the literature [30], the authorized search key is sent directly by the data owner to the data user without using the trusted centre, and the reliability of the trusted centre is stronger. Distributing the key by the trusted centre will also reduce the burden on the data owner.

This scheme realizes a searchable encryption scheme of fuzzy keywords based on blockchain in the cloud environment and the secure distribution of keys by trusted centres' the cloud server realizes the search operation and uses the blockchain's smart contract to verify the identity of its users and verify the correctness and completeness of the search results returned by the cloud server, so as to ensure the correctness of the results, realize fair payment, and store the transaction information in the blockchain to ensure the traceability of the scheme. The scenario pairs are shown in Table 3.

**Table 3.** Comparison of various schemes. Here, we use the symbol "×" to indicate that the corresponding feature is not satisfied and "√" indicates satisfied.

| Scheme | Cloud Storage | TA | Blockchain | Privacy | Identity Authentication | Correctness and Integrity Verification | Traceability | Fair Payment |
|---|---|---|---|---|---|---|---|---|
| [27] | √ | × | × | √ | × | × | × | × |
| [28] | √ | × | × | √ | × | √ | × | × |
| [29] | × | × | √ | √ | × | √ | √ | √ |
| [25] | √ | × | √ | √ | × | √ | × | √ |
| [30] | √ | × | √ | √ | √ | √ | √ | √ |
| Our scheme | √ | √ | √ | √ | √ | √ | √ | √ |

*5.3. Performance Analysis*

The results shown in Table 4 compare the computational cost of this scheme with related schemes in the key building, security index building, search token building, search phase, and verification phase. $T_M$ and $T_A$ are used to represent the execution time of a multiplication operation and addition operation, respectively, $T_H$ and $T_E$, respectively, represent the execution time of a hash operation and exponentiation operation, $T_P$; $T_F$, and $T_S$, respectively, represent the execution time of a linear pair operation, pseudo-random function operation, and signature operation, $m$ and $l$, respectively, represent the number of encryption keywords and search keywords, $n$ represents the number of documents, and $j$ represents the number of files containing keywords.

**Table 4.** A comparison of the computational cost.

| Scheme | KeyGen | Security Index Generation, Enc | Search Tokens Generation | Search Phase | Validation Phase |
|--------|--------|--------------------------------|--------------------------|--------------|------------------|
| [31] | $2T_H + 4T_M$ | $(m+4)T_M + 3mT_H + (2+m)T_P$ | $(l+2)T_M + (l+2)T_H + 2T_A + lT_P$ | $l(3T_M + 2T_H + 2T_A + 2T_P)$ | —— |
| [32] | $3T_E$ | $mT_m + 3mT_H + (2m+2)T_E$ | $3lT_H + (2l+1)T_E$ | $T_M + T_E + 3T_P$ | —— |
| [23] | $T_F + T_E$ | $mj(T_F + T_H + T_S)$ | $nT_F$ | $n(T_H + T_S)$ | $n(T_H + T_S)$ |
| [20] | —— | $5mT_H$ | $2lT_H + lT_F$ | $lT_H$ | $T_H$ |
| Our scheme | $2T_H + 4T_E$ | $2mT_F + mT_H$ | $2lT_F$ | $lT_F$ | $T_H$ |

In the key generation stage, the key generation time of this scheme is slightly longer than that of reference [32]. With the increase of the number of m, the cost of generating the security index and ciphertext will gradually increase, among which [31,32] increase the fastest, and the time of generating the search token is the same. In the search phase, the proposed scheme avoids complex computational operations, such as bilinear mapping, and has good search efficiency. The verification phase time is only one hash operation execution time, $T_H$, which is lower than the scheme in reference [20]. Moreover, the proposed scheme does not need local verification by users and uses smart contracts to hash operations, thus reducing the computational overhead of users. The computational overhead of this scheme is similar to that of reference [20] in the search and verification phase, but it costs less to generate a secure index, ciphertext, and search token. To sum up, the scheme in this paper is the best in performance.

The implementation of the key generation and file encryption part of the experiment is based on the Pairing-Based Cryptography (PBC) library, and the experimental programs are written in C and run under PBCVC on a Windows 10 operating system AMD Ryzen 7 5800H with Radeon Graphics and 16.0 GB of RAM. The class A elliptic curve provided by the PBC library is selected. SHA-256 is used for hashes, and HMAC-SHA256 is used for pseudo-random functions. The PBC library is a free C library built on the GMP library that performs the mathematical operations underlying pairing-based cryptosystems. It provides routines such as elliptic curve generation, elliptic curve arithmetic, and pairing computation. The experiments compared the computational overheads, and the results are shown in Figures 6 and 7.

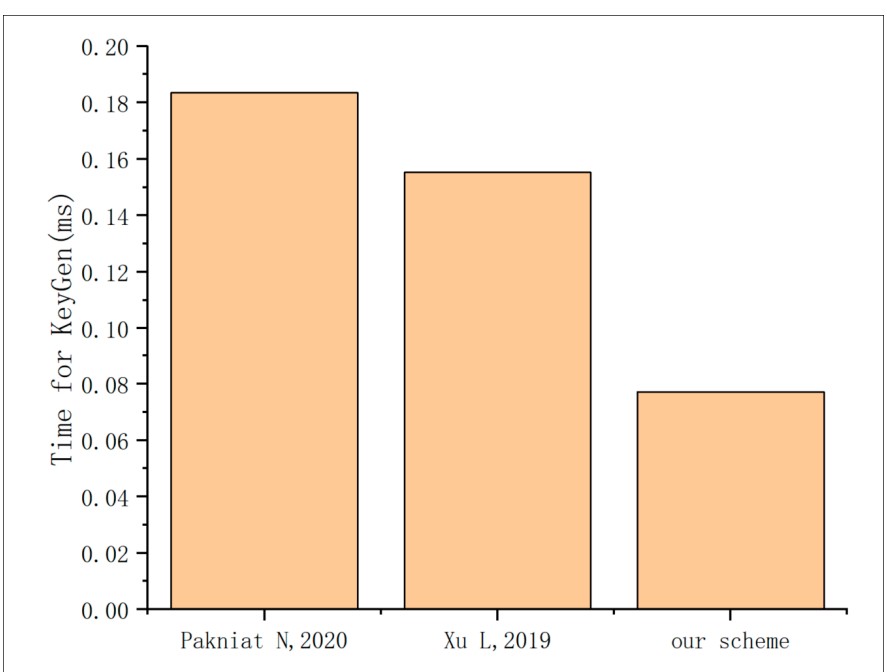

**Figure 6.** KeyGen cost comparison [31,32].

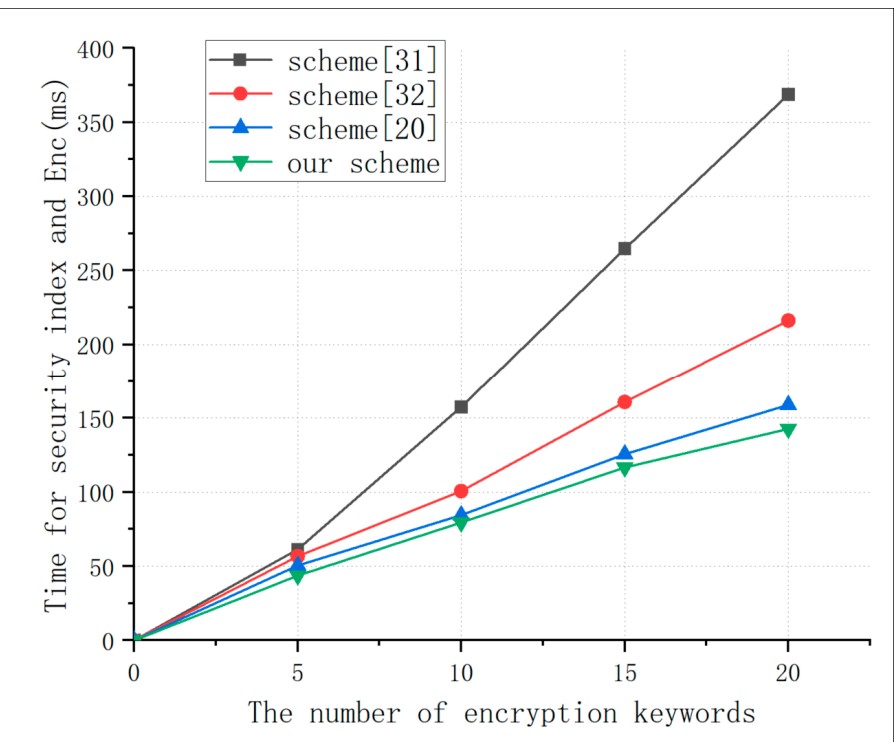

**Figure 7.** Enc cost comparison.

## 6. Conclusions

This study suggests a verifiable fuzzy keyword search encryption solution based on blockchain in a cloud context to address the dishonest conduct of hostile servers in searchable encryption schemes. The plan uses cloud servers for search operations to increase search efficiency and introduces a blockchain system, which not only stops cloud servers from purposefully returning false results and validates the accuracy and integrity of documents that are returned but also makes use of the Ethereum blockchain's payment system to safeguard the rights of cloud servers and data users and achieve fair payment.

However, the scheme cannot achieve a dynamic update of encrypted files and has limitations in practical applications, so a blockchain-based searchable encryption scheme that can be applied to dynamic cloud environments in healthcare, education, and other fields will be considered in the next step of work. Furthermore, we considered combining blockchain with verifiable computing in the new scheme to provide more secure services. As a next step, we plan to try to use formal analysis in smart contracts to reduce potential errors and costs in the contract development by using the validation methods in the formal modelling mentioned in [33,34] and the application of blockchain-based verifiable search solutions in the healthcare sector to try to address the privacy issues of medically sensitive data.

**Author Contributions:** T.F. conducted the review. B.H. made the plans and wrote the drafts. All authors have read and agreed to the published version of the manuscript.

**Funding:** This research was funded by the National Natural Science Foundation of China, grant number 61762060, and the Foundation for the Key Research and Development Program of Gansu Province, China, grant number 20YF3GA016.

**Data Availability Statement:** No data were used to support this study.

**Conflicts of Interest:** The authors declare no conflict of interest.

## Appendix A

**Theorem A1.** *This scheme satisfies the semantic security of adaptive keyword selection under the random oracle model.*

**Proof.** Suppose there is a probabilistic polynomial time, $(PPT)$ $S$, as the simulator and $A$ as the adversary. The proof equivalence from the literature [16] can be it is known that simulation-based game proofs are equivalent to indistinguishable game proofs, proving that the adversary, $A$, will win the game by analyzing the ciphertext generated by the simulator and the index and search tokens generated by the simulator to win the game. If $\left| \Pr\left[ Real_A^{\Pi}(\lambda) \right] - \Pr\left[ Ideal_{A,S}^{\Pi}(\lambda) \right] \right| \leq negl(\lambda)$, where $negl(\lambda)$ is a negligible function, then the scheme is semantically safe for an adaptive keyword selection.

We will prove that no opponent of $A$ can distinguish between $Ideal_{A,S}^{\Pi}$ and $Real_A^{\Pi}$.

The simulator generates the mock ciphertext document, $C_R'$, the mock security index, $I'$, and the mock search token, $T_{w'}'$, as follows:

- Simulate the ciphertext document, $C_R'$. From the leak function, $L_1$, the simulator inputs the document set, $D$, and generates simulated encrypted documents, $C_R' = \{C_{R1}', C_{R2}', \ldots, C_{Rn}'\}$. Because the symmetric keys $K_2, K_3$ are secure, $C_R$ and $C_R'$ are computationally indistinguishable.

$$\left| \Pr[Enc(D, K_1, K_2) \to C_R] - \Pr\left[ Random \to C_R' \right] \right| \leq negl_1(\lambda)$$

- Simulation security index, $I'$. $I'\left( \pi_{k_1}(w') \right) = Ev(w')$, where $Ev(w') = v(w') \oplus F_{k_2}\left( \pi_{k_1}(w') \right)$. In $Real_A^{\Pi}(\lambda)$, the pseudo-random function, $F$, and the pseudo-random permutation function, $\pi$, are used to construct the security index. When simulating $I'$, the random strings with the same length are used to replace the generated $\pi_{k_1}'(w')$ and $F_{k_2}'\left( \pi_{k_1}'(w') \right)$. Since the adversary, $A$, is unknown to $K_1$ and $K_2$, and the security of the pseudo-random function and the pseudo-random permutation function is known, the adversary, $A$, cannot distinguish its output from the random strings with the same length; that is, $I$ and $I'$ are indistinguishable in the calculation.

$$\left| \Pr[IndexGen(D, K_1, K_2) \to I] - \Pr\left[ Random \to I' \right] \right| \leq negl_2(\lambda)$$

- Simulated search token, $T'_{w'}$. Using the leak function, $L_2(D, w') = (F, \pi, T'_{w'})$, where $TD'_{w'} = (\pi_{k_1}(w'), F_{k_3}(\pi_{k_1}(w')))$, $F$ is a pseudo-random function, and $\pi$ is a pseudo-random permutation function (similar to 2), because $K_1$ and $K_3$ are unknown; $T_{w'}$ and $T'_{w'}$ are computationally indistinguishable.

$$\left| \Pr\left[ TokenGen(w', id_u, K_1, K_3) \to TD, T_{w'} \right] - \Pr\left[ Random \to TD'_{w'} \right] \right| \le negl_3(\lambda)$$

- Advantage of adversary $A$ $Adv_A(\lambda)$ can be divided into three parts according to the above: $Adv_A(C'_R), Adv_A(I'), Adv_A(T'_{w'})$; then,

$$Adv_A(\lambda) = Adv_A(C'_R) + Adv_A(I') + Adv_A(T'_{w'}) = \left| \Pr[Ind_A(\lambda) = 1] - \frac{1}{2} \right|$$

$$
\begin{aligned}
\Pr[Ind_A(\lambda) = 1] \quad &= \tfrac{1}{2} + Adv_A(C'_R) + Adv_A(I') + Adv_A(T'_{w'}) \\
&= \tfrac{1}{2} + \left| \Pr[Enc(D, K_1, K_2) \to C_R] - \Pr[Random \right. \\
&\quad \left. \to C'_R] \right| + \left| \Pr[IndexGen(D, K_1, K_2) \to I] - \Pr[Random \right. \\
&\quad \left. \to I'] \right| + \left| \Pr[TokenGen(w', id_u, K_1, K_3) \to TD, T_{w'}] \right. \\
&\quad \left. - \Pr[Random \to TD'_{w'}] \right| \\
&\le \tfrac{1}{2} + negl_1(\lambda) + negl_2(\lambda) + negl_3(\lambda)
\end{aligned}
$$

$$negl(\lambda) = negl_1(\lambda) + negl_2(\lambda) + negl_3(\lambda)$$

$$\Pr[Ind_A(\lambda) = 1] \le \frac{1}{2} + negl(\lambda)$$

$$\left| \Pr\left[ Real_A^\Pi(\lambda) \right] - \Pr\left[ Ideal_{A,S}^\Pi(\lambda) \right] \right| \le negl(\lambda)$$

The system meets the adaptive keyword semantic security since it is impossible to discriminate between the outputs of $Ideal_{A,S}^\Pi$ and $Real_A^\Pi$ for an adversary $A$ of any polynomial duration. □

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
