# Peer review of "Encryption Scheme of Verifiable Search Based on Blockchain in Cloud Environment"

_cryptography, doi:10.3390/cryptography7020016_

Round 1

Reviewer 1 Report

The article suggests a solution for verified fuzzy keyword search that protects privacy and is based on the Ethernet blockchain in a cloud environment and uses searchable symmetric encryption (SSE) to address the security concerns mentioned above. Using the immutability of the Ethernet blockchain system, the search user can check the accuracy and integrity of the query document to make sure the cloud server doesn't send back wrong results. This method also uses smart contracts on the Ethernet blockchain to make sure that the cloud server and the person who wants the data trade fairly.

  Comments and Suggestions:  

- The article covers an interesting topic, and the obtained results are promising.  

- The abstract is a bit short and needs to be extended.  

- In addition, the authors need to insert a short paragraph at the end of the introduction which describes the structure of the paper.  

- The authors may also insert a new figure between the first and second sections that illustrates graphically the proposed approach.  

- The related section has to be summarized in tabular form.  

- In order to emphasize the originality of their contribution, the authors must identify the limitations of existing related works.  

- The authors are invited to include a short paragraph about the use of formal methods for the verification of smart contracts.

- For this purpose, the authors are invited to consider the following interesting reference (and others):
1. https://ieeexplore.ieee.org/document/9970534

2. https://www.sciencedirect.com/science/article/abs/pii/S1574119220300821   

- Section 3.2. is made of very short paragraphs. The authors need to avoid this everywhere.  

- Line 170: "In this paper, we use the security model of the literature [23]," ===> The authors are invited to argue more about the choice of the adopted security model.  

- Are there other security models that can be used? Please provide comparisons with other possible models.  

-  Line 257: "deleted on the basis of literature [24]," ===> please provide more details about the adopted technique.  

- Figure 2 is not clear enough and needs some improvement.  

- Is it possible to give an example of adopted/proposed smart contracts?  

- Is it also possible to share the code of the developed solution?  

- It is preferable to move the proof of Theorem 1 to the appendix.  

- The authors must identify the limitations of their work and suggest additional future work directions.  

- The conclusion is too short and needs to be extended.

Author Response

请参阅附件。

Reviewer 2 Report

This manuscript is proposing a blockchain-based encryption scheme of verifiable search. A few commands that I would like to make for possibly improving the work.

1. The motivation of the paper should be made it more clear. Are authors try to improve the accuracy of the search results or privacy protection? It seems that the proposed scheme requires user's identity information and transaction records to be saved on the blockchain. 

2. The authors in Table 2 and Table 3 compare the the proposed method to many other methods in the literature. Is it possible to conduct some numerical experiments more convincing? 

Reviewer 3 Report

Regarding the content, I do not have any changes to recommend, it makes a good literary review to support the relevance of the problem to be studied and a good structuring of the content, it uses the correct methodology for this type of study and it is a consistent and well-detailed methodology to give significance to the results they show, makes a good discussion of the results with respect to the studies carried out previously, and marks the conclusion obtained well.

Although I advise looking at these things:

Never two sections without a paragraph of text in between. You should put a couple of lines describing/naming the subsections you are going to deal with within that section. This should be corrected between sections 5-5.1.

In the section “6. Conclusions”, it is necessary to develop a deeper analysis of the conclusions, implications and limitations of the study. In addition to the possible future lines of research opened with this research.

And the references in the 'References' section must follow the model set by the journal. You must correct the errors that exist. Look at this in the template.

Round 2

Reviewer 1 Report

1. Figure 1 needs some improvement and some expalnation about it need to be provided 

2. Table 1: Please add a column which indicates the year of publication of each article. The contributions and limitaitons need to be presented in a better way.

3. Section 3.2.: Please avoid the use of short paragraphs in this section and everywhere in the paper.

4. Conlusion: "we plan to experiment with the use of formal analytics for smart contracts" ===> authors need to add some references about this notion of fomal analytics

5. For this purpose they may include the following references (and others):

a. https://ieeexplore.ieee.org/abstract/document/9970534

b. https://dl.acm.org/doi/abs/10.1145/3185089.3185138

6. Line 445: "based on the Pairing-BasedCryptography (PBC) library" ==> Please provide more details about this library and share the link from which it may be downloaded.

7. The appendix must be placed after the list of references.

Reviewer 2 Report

I do not have any additional comments.

Author Response

Thank you for your review and your constructive comments on the article.